# Safety Profile of a Virus-Like Particle-Based Vaccine Targeting Self-Protein Interleukin-5 in Horses

**DOI:** 10.3390/vaccines8020213

**Published:** 2020-05-09

**Authors:** Sigridur Jonsdottir, Victoria Fettelschoss, Florian Olomski, Stephanie C. Talker, Jelena Mirkovitch, Tanya Rhiner, Katharina Birkmann, Franziska Thoms, Bettina Wagner, Martin F. Bachmann, Thomas M. Kündig, Eliane Marti, Antonia Fettelschoss-Gabriel

**Affiliations:** 1Clinical Immunology Group, Department for Clinical Research VPH, Vetsuisse Faculty of the University of Bern, Länggassstrasse 124, 3012 Bern, Switzerland; sigridur.jonsdottir@vetsuisse.unibe.ch (S.J.); jelena.mirkovitch@vetsuisse.unibe.ch (J.M.); eliane.marti@vetsuisse.unibe.ch (E.M.); 2Department of Dermatology, University Hospital Zurich, Wagistrasse 12, 8952 Schlieren, Switzerland; victoria.fettelschoss@usz.ch (V.F.); florian.olomski@usz.ch (F.O.); tanya.rhiner@uzh.ch (T.R.); franziskazabel@hotmail.com (F.T.); 3Faculty of Medicine, University of Zurich, 8091 Zurich, Switzerland; 4Evax AG, Hörnlistrass 3, 9542 Münchwilen, Switzerland; katharina@evax.ch; 5Institute of Virology and Immunology, Länggassstrasse 122, 3012 Bern, Switzerland; stephanie.talker@vetsuisse.unibe.ch; 6Department of Infectious Diseases and Pathobiology, Vetsuisse Faculty, University of Bern, Länggassstrasse 122, 3012 Bern, Switzerland; 7Departments of Population Medicine and Diagnostic Sciences, College of Veterinary Medicine, Cornell University, Ithaca, NY 14853-0001, USA; bw73@cornell.edu; 8RIA Immunology, Inselspital, University of Bern, 3012 Bern, Switzerland; Martin.Bachmann@insel.ch; 9Jenner Institute, Nuffield Department of Medicine, Henry Welcome Building for Molecular Physiology, University of Oxford, OX1 2JD Oxford, UK; 10Department of Dermatology, University Hospital Zurich, Gloriastrasse 31, 8091 Zurich, Switzerland; Thomas.kuendig@usz.ch

**Keywords:** therapeutic vaccine, self-protein, safety

## Abstract

*Background:* Insect bite hypersensitivity (IBH) is an eosinophilic allergic dermatitis of horses caused by type I/IVb reactions against mainly *Culicoides* bites. The vaccination of IBH-affected horses with equine IL-5 coupled to the Cucumber mosaic virus-like particle (eIL-5-CuMV_TT_) induces IL-5-specific auto-antibodies, resulting in a significant reduction in eosinophil levels in blood and clinical signs. *Objective:* the preclinical and clinical safety of the eIL-5-CuMV_TT_ vaccine. *Methods:* The B cell responses were assessed by longitudinal measurement of IL-5- and CuMV_TT_-specific IgG in the serum and plasma of vaccinated and unvaccinated horses. Further, peripheral blood mononuclear cells (PBMCs) from the same horses were re-stimulated in vitro for the proliferation and IFN-γ production of specific T cells. In addition, we evaluated longitudinal kidney and liver parameters and the general blood status. An endogenous protein challenge was performed in murine IL-5-vaccinated mice. Results: The vaccine was well tolerated as assessed by serum and cellular biomarkers and also induced reversible and neutralizing antibody titers in horses and mice. Endogenous IL-5 stimulation was unable to re-induce anti-IL-5 production. The CD4^+^ T cells of vaccinated horses produced significantly more IFN-γ and showed a stronger proliferation following stimulation with CuMV_TT_ as compared to the unvaccinated controls. Re-stimulation using *E. coli*-derived proteins induced low levels of IFNγ^+^CD4^+^ cells in vaccinated horses; however, no IFN-γ and proliferation were induced following the HEK-eIL-5 re-stimulation. *Conclusions:* Vaccination using eIL-5-CuMV_TT_ induces a strong B-cell as well as CuMV_TT_-specific T cell response without the induction of IL-5-specific T cell responses. Hence, B-cell unresponsiveness against self-IL-5 can be bypassed by inducing CuMV_TT_ carrier-specific T cells, making the vaccine a safe therapeutic option for IBH-affected horses.

## 1. Introduction

Insect bite hypersensitivity (IBH) is a type I/IVb hypersensitivity to mainly *Culicoides* salivary gland proteins characterized by the induction of IgE, activation of mast cells and basophils and late-phase eosinophil accumulation in the skin [1,2]. The key molecule in the differentiation, activation, and survival of eosinophils is Interleukin-5 (IL-5) [3,4,5,6]. We recently presented IL-5 to be a suitable target for treating IBH-affected horses and developed a therapeutic vaccine consisting of equine IL-5 (eIL-5) linked to virus-like particles (VLPs). The vaccine induces IL-5-specific auto-antibodies, thereby significantly reducing the levels of eosinophils and clinical signs of IBH [7,8]. VLPs are supra-molecular structures consisting of one or more structural viral proteins, but lacking the genetic information of a virus, representing an important safety aspect of VLPs. The immune system strongly responds to VLPs due to their key immunological features such as their particulate structure, size, repetitive surface, and pathogen-associated molecular patterns (PAMPs), which makes VLPs a useful tool for vaccine development [9,10,11]. Coupling a non-immunogenic soluble self-protein to a highly immunogenic VLP forms hapten-carrier-like conjugates. In that way, the non-immunogenic hapten part becomes “visible” to the immune system and receives T cell help as a bystander signal derived from the foreign immunogenic VLP carrier [12,13,14]. As a result, T cell-dependent long-lasting IgG antibody responses are developed against both the carrier VLP and the hapten self-molecule, although hapten-specific T cell responses are absent. In the case of a hapten being a self-molecule, it is of particular importance that no auto-reactive T cells are generated upon vaccination, thus protecting from auto-immunity. In contrast, self-reactive B cells are less problematic, as B cells are tightly regulated by T cells and the latter are strictly selected by positive and negative selections during thymic T cell repertoire development, creating a self-tolerant T cell repertoire [15,16]. Taken together, safety aspects of vaccines targeting self-molecules demand different safety requirements for B and T cells. Important for the control of auto-reactive B cell responses are the reversibility of the IgG antibody response and the lack of their activation through endogenous protein other than by the vaccine. For T cells, safety requirements for vaccination against self-proteins are stricter, as the vaccine should not induce any auto-reactive T cells. Here, we want to shed light on the B and T cell safety of anti-self vaccines including various other clinical safety parameters in horses being vaccinated against the self-protein IL-5 for several consecutive years using the eIL-5-CuMV_TT_ vaccine.

## 2. Materials and Methods

### 2.1. Mice

Female C57BL/6 mice were purchased from Harlan, Horst. The mice were kept under specified pathogen-free conditions. All the animal experimentation was approved by the Swiss cantonal veterinary authorities.

### 2.2. MIL-5-Qβ Vaccine

The cloning, expression, purification, and vaccine production of mouse IL-5 (mIL-5) and Qβ was previously described [17].

### 2.3. Endogenous Challenge in Mice

The mice were subcutaneously immunized three times in bi-weekly intervals (days 0, 14, and 28) using mIL-5-Qβ. Two weeks after the last vaccination, the mice were intravenously challenged using either the vaccine or the mIL-5 protein alone or PBS (Gibco, Thermo Fisher Scientific, NY, USA) as control (day 42). The bleeding was performed at baseline, one day prior to the challenge and two weeks after the challenge (day 0, 41, and 56).

### 2.4. MIL-5-Specific ELISA

The IL-5-specific antibodies were semi-quantified by ELISA in serum on day 0, 41, and 56, as previously described [17]. However, using mIL-5 produced by HEK cells (NRC CNRC, Montreal, Canada), HEK-mIL-5 (see below section ‘recombinant proteins for stimulation and ELISA’). The OD_50_ was calculated for evaluation of the antibody titer.

### 2.5. Horses and Vaccination

All the study horses were client-owned IBH-affected horses and participated in the clinical studies of Evax AG (Switzerland) and therefore had been vaccinated with eIL-5-CuMV_TT_ in accordance with the previously published clinical studies [7,8] (Appendix A). All the horse owners signed informed consent. All the animal experimentation was approved by Swiss cantonal veterinary authorities, ethical approval code 25152, 28711, 29065, 27530, 29968. The vaccine was produced as described earlier [7].

In 2018 and 2020, blood samples were collected from 34 Icelandic horses that participated in two clinical trials starting in 2015 (group 1,2) or 2017 (group 3,4) and had received the active vaccine for the 1st (8 horses), 2nd (9 horses), 3rd (11 horses), 4th (4 horses), or 5th (2 horses) year, respectively. Data of the 3rd, 4th, and 5th year-vaccinated horses were combined into 3–5 year samples. Control blood was collected from unvaccinated Icelandic horses, i.e., 7 placebo-vaccinated IBH-affected and 9 non-vaccinated healthy horses. IFN-γ production: All the above-described horses were included into an intracellular IFN-γ measurement of the re-stimulated equine peripheral blood mononuclear cells (PBMCs). However, six of these horses with an active vaccine for the 3rd (2 horses), 4th (3 horses), or 5th (1 horse) year had to be excluded from the common analysis due to the high assay background already in the medium control samples, most likely due to assay medium contamination in all samples from these horses. Unfortunately, the experiment could not be repeated due to the shutdown of the university’s flow cytometry facility during the corona crisis. Therefore, these six horses are shown separately (CD4^+^
Figure 1b; CD8^+^
Figure 1d) from the other horses (CD4^+^
Figure 1a; CD8^+^
Figure 1c), and in order to correct for the background, a quotient of the stimulant divided by the medium control values is shown instead. Proliferation: In total, there were 21 horses, i.e., 10 unvaccinated (placebo) control horses and 11 vaccinated horses that received an active vaccine for the 1st (5 horses), 2nd (4 horses), or 3rd (2 horses) year. Also, due to the shutdown of the university laboratories including flow cytometry facility, we could not include any more horses that were planned for the measurement in 2020.

### 2.6. Recombinant Proteins for Stimulation and ELISA

*E. coli*-eIL-5 and CuMV_TT_ were produced in the *E. coli* strain C2566 (New England Biolabs, MA, USA) and purified and refolded, as previously described [7,8,18]. HEK-eIL-5 or HEK-mIL-5 are mature eIL-5 (mature Interleukin-5, equus caballus; UniProt O02699) or mature mIL-5 (mature Interleukin-5, mus musculus; UniProt P04401), respectively, and were in silico codon optimized, N-terminally connected to a mouse Igκ signal sequence, and C-terminally connected to a GGC-linker followed by a Strep-tag and a stop codon. The resulting fragment was synthesized (IDT Genomics, USA) and integrated into the expression vector pCB14 by isothermal cloning. The HEK 293T cells were grown to an 80% confluency and transfected with polyethylenimine (PEI) (PEI Max 40K, Polysciences, USA) at a PEI to DNA ratio of 2.5. The eIL-5 and mIL-5-containing supernatants were harvested after 5 days and purified via the Strep-tag by Strep-Tactin (IBA Lifesciences, Göttingen, Germany) column, respectively. Murine Granulocyte-macrophage colony-stimulating factor (GM-CSF) is mature mGM-CSF (mature Granulocyte-macrophage colony-stimulating factor, mus musculus; UniProt P01587) and was codon-optimized and flanked with 5′NdeI and 3′XhoI restriction sites in silico. The fragment was synthesized (IDT Genomics, USA) and integrated into the expression vector pET42b(+), resulting in mGM-CSF, which is followed by a hexa His-tag and an in-frame stop codon and was expressed in the *E. coli* strain BL21DE03 (New England Biolabs, MA, USA). The overnight cultures were grown at 37 °C and diluted with LB medium containing 50 mg/L of kanamycin. The cultures were induced by adding IPTG to a final concentration of 1 mM at OD_600_ values of 0.6, followed by 4 h incubation. The culture was centrifuged and the resulting pellet re-suspended in 2 mM EDTA, 0.5% Triton X-100, 50 mM TrisHCl, pH 8.0. Inclusion bodies were prepared and solubilized in 6 M GdmCl, 10 mM DTT, 50 mM TrisHCl, pH 8.0. The murine GM-CSF was purified via the His-tag by a NiNTA (NiNTA Superflow, Qiagen, MD, USA) column and eluted with an elution buffer 6 M GdmCl, 0.5 M NaCl, 0.5 M Imidazole, 20 mM TrisHCl, pH 5.9. The eluate was diluted with the elution buffer to 1 mg/mL, mixed 1:1 with 2 M GdmCl, 50 mM Tris, pH 8.0 and subsequently dialyzed.

### 2.7. Isolation of Equine PBMCs

Blood was collected from the jugular vein in heparin-tubes (Greiner) two weeks after the annual boost. The peripheral blood mononuclear cells (PBMCs) were isolated by the density gradient using Biocoll, as described previously [19]. The PBMCs were cultured in RPMI (Thermo Fisher Scientific, NY, USA), supplemented with L-Glutamine, HEPES, 10% inactivated horse serum or FBS (Merck, Germany), 100 U/mL Penicillin, 100 μg/mL Streptomycin (Merck), 1mM Sodium pyruvate (Merck), 1% non-essential amino acids (Merck), MEM vitamins (Merck), and 2-Mercaptoethanol (0.05 mM).

### 2.8. Intracellular IFN-γ Measurement of Re-Stimulated Equine PBMCs

The 5 × 10^6^ cells/well-isolated PBMCs were seeded in 12-well plates. For the quantification of the intracellular IFN-γ of the CD4^+^ and CD8^+^ T cells, the cells were stimulated for 18 h at 37 °C with 1μg/mL *E. coli*-eIL-5, HEK-eIL-5, CuMV_TT_, Tetanus Toxoid (Astarte Biologics, WA, USA) as a positive control for the antigen-specific response, or *E. coli*-mGM-CSF as an irrelevant protein stimulus. Alternatively, the cells were left unstimulated as a negative control. Of note, not all stimuli were tested for all horses. After 14 h stimulation, Brefeldin A (Sigma, MO, USA) (10 μg/mL) was added to all wells and phorbol 12-myristate 13-acetate (Sigma, MO, USA) (PMA; 25 ng/mL) and ionomycin (Sigma, MO, USA) (I; 1 μg/mL) were added to one well with unstimulated cells as a positive control (data not shown). Four hours later, the cells were harvested and washed (centrifugation for 10 min at 600xg at 4 °C). Staining was performed with mouse anti equine CD4-FITC (1:40, clone CVS4, Biorad, USA) and mouse anti-equine CD8β-PE (1:10, clone HT14A, Washington State University, WA, USA), then they were labeled with the Zenon R-Phycoerythrin mouse IgG1 labeling kit (Thermo Fisher Scientific, NY, USA) and the LIVE/DEAD^TM^ Fixable Near-IR dead-cell stain kit (1:300, Thermo Fisher Scientific, NY, USA) for 15 min at 4 °C. Cells were washed, fixed and permeabilized by Fixation/Permeabilization solution (BD Biosciences, Allschwil, Switzerland) for 30 min at 4 °C. Cells were washed with 1 × BD Perm/Wash and stained with mouse anti-equine IFN-γ-Alexa 647 (1:500, clone 38-1, Wagner Laboratory, Cornell University, USA) for 30 min at 4 °C. The cells were acquired with a FACS Canto II flow cytometer (BD Biosciences, Allschwil, Switzerland). The compensation was calculated by the FACSDiva software with single-stained compensation controls. The gating strategy is shown for the tetanus toxoid stimulation (Appendix A). The results are shown as percentages of CD4^+^CD8^-^ or CD8^+^CD4^-^ cells producing IFN-γ following re-stimulation with different stimulants.

### 2.9. Cell Proliferation of Re-Stimulated Equine PBMCs

Isolated PBMCs were stained with the CellTrace^TM^ violet cell proliferation kit (Thermo Fisher Scientific, USA) for the detection of proliferated cells following 7 days of re-stimulation. Up to 40 × 10^6^ PBMCs were stained according to the manufacturer’s instructions; 2 × 10^6^ stained PBMCs in 1 mL medium/well were seeded in 24-well plates. The cells were stimulated for 7 days at 37 °C with 1μg/mL of *E. coli*-eIL-5, HEK-eIL-5, CuMV_TT_, Tetanus Toxoid (NiNTA Superflow, Qiagen, MD, USA)), *E. coli*-mGM-CSF, or ConA (Sigma, MO, USA) (data not shown), or left unstimulated as a negative control. Of note, not all stimuli were tested for all horses. At day 7, the cells were harvested and stained in one step with mouse anti-equine CD4-FITC, mouse anti-equine CD8β-PE, and the LIVE/DEAD Near IR, as described. The gating strategy is shown for tetanus toxoid stimulation (Appendix A). The results are shown as the percentage of proliferating CD4^+^CD8^-^ or CD8^+^CD4^-^ following re-stimulation with different stimulants.

### 2.10. Stimulation of Equine PBMCs with Resiquimod and Detection of Specific IgG

The isolated PBMCs (1 × 10^6^ cells/well) were seeded in 1 mL complete medium in 24-well plates. The cells were stimulated with 5 μg/mL Resiquimod (Sigma, MO, USA) [20], cells were incubated for 7 days at 37 °C and the supernatant was collected for IgG detection.

### 2.11. IL-5 and CuMV_TT_ Specific IgG ELISA

An antigen-specific IgG ELISA was carried out, as described by Jonsdottir et al. [21], except that the plates were coated with *E. coli*-eIL-5, HEK-eIL-5 or CuMV_TT_. The IgG subclasses, IgG1, IgG4/7, and IgG5, were measured in plasma and sera at a 1:100 dilution, whereas the total IgG was measured in the undiluted supernatant of PBMCs. The IgG subclass-specific antibodies were previously described [22]. The quantification of total antigen-specific IgG in sera was described by Fettelschoss-Gabriel et al. [8]. Additionally, the supernatant was examined for the total IgG against Tetanus Toxoid (Schweizerisches Serum und Impfinstitut, Bern, Switzerland, [23]).

### 2.12. Avidity of IL-5-Specific IgG

For an avidity evaluation of IL-5-specific IgG after the yearly booster of the 1st, 2nd, and 3rd year, maxisorp 96-well ELISA plates (Nunc) were coated with purified *E. coli*-eIL-5 (5 mg/L) in a coating buffer (0.1 M NaHCO_3,_ pH 9.6) overnight at 4 °C. The plates were blocked with Superblock (Thermo Fisher Scientific, NY, USA) for 2 h at RT. Two sets of pre-diluted 1:10 sera were transferred to the ELISA plates and further serially diluted (three-fold dilutions) in Superblock. After 2 h incubation at RT, the sera were washed off and the plates washed 3 times for 5 min either with 7 M urea in PBST (PBS + 0.05% tween 20) or PBST only. All the plates were incubated with HRP-conjugated anti-equine IgG (1:2000, Jackson ImmunoResearch, PA, USA) at RT for 30 min before being developed with TMB (Thermo Scientific, USA) for 1 min. The absorbance was measured at 450 nm (OD_450_) on a Tecan Spark 10M spectrophotometer (Tecan, Grödig, Austria). The OD_50_ titer is described as the reciprocal serum dilution, where OD_450_ reaches the half maximum OD [8]. Titers below 10 (and including 10) were considered as background. The avidity data are described as the avidity index (AI) and refer to the ratio of the same dilution treated with and without 7 M urea (AI = OD (dilution x) + urea/OD (dilution x) no urea).

### 2.13. Clinical Chemistry and Hematology Analysis

The liver and kidney parameters were monitored in the serum of clinical study horses to observe the changes upon vaccinations. A differential blood analysis was measured by EDTA blood. The parameters were assessed in the placebo year; the 1st, 2nd, and 3rd treatment year; before the first vaccination (in January, February, or March; S = spring); and at the season end (September; A = autumn). The serum and blood were analyzed by IDEXX Diavet, Freienbach, Switzerland.

### 2.14. Statistical Analysis

The statistical analysis and preparation of graphs were performed with GraphPad Prism 7. A Shapiro–Wilk normality test showed that the data sets were not normally distributed (data not shown), and therefore we used non-parametric tests. The statistical tests were performed as indicated in the Figures. A statistical result is only indicated when statistically significant. Considered to be statistically significant were *p*-values lower than 0.05: * *p* < 0.05; ***p* < 0.01; ****p* < 0.001; *****p* < 0.0001.

## 3. Results

### 3.1. No Induction of Auto-Reactive Peripheral Blood T Cells Upon Vaccination

To investigate if a long-term vaccination with eIL-5-CuMV_TT_ induces IL-5 reactive T cells, PBMCs from horses vaccinated for 1, 2, and 3–5 years (vaccination and bleeding regimen in Appendix A) and from unvaccinated controls were re-stimulated in vitro with *E. coli*-eIL-5, HEK-eIL-5, and the VLP CuMV_TT_. In addition, the cells were also treated with tetanus toxoid as a positive control for antigen-specific T cell responses, as all the horses were previously vaccinated against tetanus by their owners and also with an unrelated *E. coli* expressed *E. coli*-mGM-CSF as an irrelevant protein. The T cell activation was quantified by the frequency of intracellular IFN-γ-positive CD4^+^ and CD8^+^ T cells following treatment with different stimulants. In addition, the proliferation of CD4^+^ and CD8^+^ T cells of the same PBMCs from part of the horses were treated with the same stimulants.

#### 3.1.1. IFN-γ Production

The IFN-γ production following any stimulation was compared to the control of unstimulated cells (medium) and vaccinated to unvaccinated horses within the stimulant. The frequency of IFN-γ-producing CD4^+^ T cells was significantly increased within PBMCs from vaccinated horses following stimulation with CuMV_TT_ and the tetanus toxoid, whereas only for tetanus toxoid unvaccinated horses showed a significantly increased IFN-γ frequency. Of note, the response to the tetanus toxoid varied due to varying time points for the tetanus vaccination. For stimulation with the unrelated *E. coli*-produced mGM-CSF and *E. coli*-mGM-CSF, we found a small but significantly increased frequency of IFN-γ-producing CD4^+^ T cells when comparing the *E. coli*-mGM-CSF and medium-stimulated vaccinated horses. This suggested that *E. coli*-derived traces within the purified protein might lead to the unspecific re-stimulation of PBMCs. In line with this, the re-stimulation with *E. coli*-produced CuMV_TT_ led to significantly increased IFN-γ frequencies in unvaccinated horses comparing the medium and CuMV_TT_ re-stimulation. Therefore, we performed re-stimulation of eIL-5 not only with *E. coli*-eIL-5 but also with HEK-eIL-5. As expected, *E. coli*-eIL-5 led to a statistically significantly increased frequency of IFN-γ-producing CD4^+^ T cells, while there was no difference for HEK-eIL-5 re-stimulation either when comparing to medium or comparing the vaccinated and unvaccinated (Figure 1a). Six horses from the 3–5 years group were analyzed separately due to the medium control background, and they showed comparable results (Figure 1b), nonetheless reflecting the results in Figure 1a nicely. Furthermore, the treatment duration and numbers of vaccine injections, either in the 1st vaccination year after the last booster (blue circles), 2nd (red circles), or 3–5 (green) vaccination years, did not influence the frequency of IFN-γ-producing CD4^+^ T cells upon HEK-eIL-5 re-stimulation or any other stimulants (Figure 1a,b).

The difference between the vaccinated and control horses was less obvious for the CD8^+^ T cells, but in general correlated well with the frequencies observed for the CD4^+^ T cells (Figure 1c). Comparably, the six separately analyzed horses also showed similar results (Figure 1d), and also no influence of vaccination duration was found (Figure 1c,d).

**Figure 1 vaccines-08-00213-f001:**
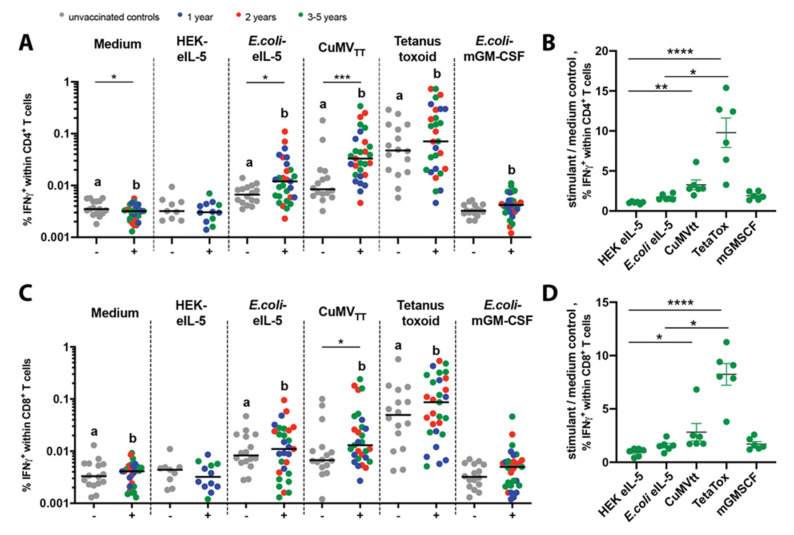
No auto-reactive CD4^+^ and CD8+ T cells upon the in vitro re-stimulation of peripheral blood mononuclear cells (PBMCs) from the vaccinated and control horses. In vitro re-stimulation using the medium control, HEK-eIL-5, *E. coli*-eIL-5, CuMV_TT_, tetanus toxoid (TetaTox), or *E. coli*-mGM-CSF of PBMCs from unvaccinated (−) control horses (grey) and 1st (blue), 2nd (red), or 3–5 (green) year vaccinated (+) horses. (**A**,**B**) CD4^+^ T cells, percentages of IFNγ-producing cells (**A**), percentages of IFNγ-producing cells of quotient stimuli/medium (**B**). (**C**,**D**) CD8^+^ T cells, percentages of IFNγ-producing cells (**C**), percentages of IFNγ-producing cells of quotient stimuli/medium (**D**). Values of individual horses with median on a log scale. Differences between the unvaccinated (−) controls and vaccinated (+) horses were analyzed with the Mann-Whitney U test. The medium unstimulated cells were compared to the stimulated cells by the Wilcoxon matched pairs rank with Bonferroni correction. There are significant differences between the unstimulated and stimulated cells within groups indicated with a for the controls and b for vaccinated horses. Medium corrected samples were analyzed by Kruskal–Wallis with Dunn’s multiple comparison. * *p* < 0.05; ** *p* < 0.01; *** *p* < 0.001; **** *p* < 0.0001.

#### 3.1.2. Proliferation

Similar results were obtained as well for the proliferation of CD4^+^ (Appendix A) and CD8^+^ (Appendix A) T cells, thus reflecting the IFN-γ-results.

*IgG.* In parallel to the T cell re-stimulation experiments, a fraction of the PBMCs were stimulated with the TLR-7 agonist resiquimod in order to show the potential reactivity of the PBMCs and to demonstrate the relevance of HEK-eIL-5. The specific IgG was measured in the cell supernatant by ELISA and indeed not only *E. coli*-eIL-5 (Appendix A) but also HEK-eIL-5 (Appendix A) showed increased levels of IL-5-specific IgG antibodies in the vaccinated horses (blue, 1 year) when compared to the unvaccinated control (grey). Moreover, the HEK-eIL-5-specific IgG antibody determination led to a lower variability in the unvaccinated horses. CuMV_TT_-specific IgG were also found to be significantly increased in the vaccinated horses (Appendix A). In contrast, the tetanus-specific IgG did not differ between the two groups (Appendix A).

### 3.2. Anti-Self Antibody Induction Is Robust, Reversible, Neutralizing, and Not Auto-Induced

To assess the potential of endogenous IL-5 to boost the previously established vaccine-induced IL-5-specific antibody responses and thus have the risk for uncontrolled anti-self antibody production, a challenge experiment in mice was conducted. To this end, the mice received three immunizations of mIL-5-VLP to establish anti-IL-5 antibody titers. Two weeks after the third vaccination, the mice were challenged either with soluble mIL-5 mimicking the endogenous protein, with mIL-5-VLP as a positive control and PBS as a negative control. Serum was collected before vaccination, prior to the challenge, and two weeks post challenge in order to compare the IL-5-specific titers which were measured by ELISA.

The vaccination with mIL-5-VLP induced strong mIL-5 antibody titers in all animals detected on day 41. As expected, mice that were challenged with the mIL-5-VLP significantly increased their anti-mIL-5 antibody titers detected at day 56, whereas the mice challenged with PBS or with soluble mIL-5 alone showed no increase in mIL-5-specific titers (Figure 2a).

To test for the reversibility of the specific antibody responses in horses, anti-eIL-5 IgG antibody titers were followed before and after eIL-5-CuMV_TT_ immunizations in three consecutive years during several time points throughout the study, with the final time point prior to the annual booster of the fourth year. The vaccination regimen is shown in Appendix A.

In the first year, anti-IL-5 IgG antibody titers were successfully induced upon two vaccinations and a mid-season booster injection and dropped towards the end of the season. In the consecutive two years, a single booster was sufficient to raise the anti-IL-5 antibody titers. Of note, the specific IgG antibody levels almost returned to pre-immune levels after each season. The peak antibody titers measured after the booster injections in the following years were found to achieve levels of specific IgG close to the peak titers observed after two initial vaccinations. The serological analysis showed that the specific antibody responses were reversible within one year and could be efficiently boosted once a year by a single injection (Figure 2b).

In order to show that the anti-IL-5 antibody production in all vaccination years was equally efficient, we show in the same horses the eosinophil levels in the blood in the placebo-treated year and first and second treatment year. When compared to the placebo year, equally significantly reduced eosinophil levels in the blood were seen in the first vaccination year, with a total of three injections, and in the second vaccination year with the single booster (Figure 2c).

We further analyzed the total IgG antibodies and determined the avidity of IL-5-specific IgG in eight horses after the first, second, and third year by performing a titration ELISA in the presence and absence of 7 M Urea as the chaotropic agent. The treatment by urea dissociated all the low avidity antibodies, whereas the high avidity antibodies remain bound. The avidity index indicated a trend towards an increase in avidity in the second vaccination year compared to the first year, which was maintained in the third year (Figure 2d).

### 3.3. IgG Antibody Subclasses

In parallel to the PBMCs stimulation experiments, IL-5- and CuMV_TT_-specific IgG subclasses were assessed in plasma.

The IL-5- and CuMV_TT_-specific antibodies induced by the vaccine were mostly of the subclass IgG1 and IgG4/7 and, to a lower extent, IgG5. All the vaccinated horses produced significant levels of IgG1, IgG4/7, and IgG5 against *E. coli*-eIL-5 (Figure 3a), HEK-eIL-5 (Figure 3b), and CuMV_TT_ (Figure 3c) in the plasma as compared to the controls. The non-vaccinated control horses did not show any meaningful antibody level tested for *E. coli*-, HEK-IL-5, or CuMV_TT_.

As IgG1 and IgG4/7 were found to be the main IgG subclasses induced by vaccination with the eIL-5-CuMV_TT_ assessed in plasma, the IL-5- and CuMV_TT_-specific IgG1 and IgG4/7 were measured in serum at different time points: 1st year before vaccination (V), after the 2nd vaccination (A), when boosted (V1), and after the boost (A1); 2nd year when boosted (V2), and after the boost (A2); and 3rd year when boosted (V3), and after the boost (A3). CuMV_TT_- (Figure 3d) and HEK-IL-5-specific (Figure 3e) IgG1 and IgG4/7 are shown. After the two initial vaccinations (V), the horses had high levels of CuMV_TT_- and IL-5-specific IgG1 and IgG4/7. From the time of the 2nd vaccination until the boost, the IL-5-specific IgG1 levels decreased more than the IgG4/7 levels. For CuMV_TT_, the surrogate marker, the annual boost led to similar levels of CuMV_TT_-specific IgG1 and IgG4/7 as after the initial vaccinations. In contrast, the levels of IL-5-specific IgG1 and IgG4/7 seemed to decline over time.

### 3.4. No Induction of Immune Complex Disease and Good General Health Status

IL-5-specific antibodies bind to their target protein IL-5 and thus can form immune complexes (ICs) in the blood. During steady state, ICs are cleared by erythrocytes or platelets (dependent on the species) in the liver and spleen via the complement system in order to prevent depositions in the circulation that could damage the kidney [24]. For the six liver and nine kidney parameters measured, no deviation was observed over three years of vaccination compared to the normal values observed in the placebo-treated horses. Over the course of the three-year study, we observed only two outliers—i.e., in one horse for the Serum Amyloid A (SAA) value (most likely the horse had an acute infection), and in one horse with increased symmetric dimethylarginine (SDMA). Both values were within the normal range in a repeated measurement two to four weeks later (Figure 4 and Appendix A).

In addition to the clinical chemistry assessment, a blood differential analysis followed the numbers of the leukocyte, neutrophil, and erythrocyte counts as well as the hematocrit (%) and hemoglobin (mmol/L). There were no meaningful differences observed at any time after vaccination compared to the baseline or placebo treatment (Appendix A).

## 4. Discussion

Vaccinating IBH-affected horses with eIL-5-CuMV_TT_ was shown to be an effective way to improve the clinical symptoms of the disease [7,8]. Here we present data that support the safety of the IL-5 vaccine and allow drawing conclusions regarding the tolerability and toxicity of the vaccine, which induced IL-5-specific IgG antibodies without us observing any adverse events throughout the longitudinal study. The therapeutic vaccine aims at the activation of IL-5-specific B cells to produce antibodies that neutralize endogenous IL-5 and subsequently reduce numbers of eosinophils, promoting clinical benefit of IBH-related symptoms in horses. In line with previously published data in mice on a VLP-based vaccine targeting IL-1β in a model for type two diabetes [25], our vaccine does not induce IL-5-specific T cells in horses, whereas the IL-5-specific B cells do receive bystander T cell help from VLP-specific T cells in order to generate long-lived and reversible anti-IL-5 IgG antibodies.

B cell unresponsiveness towards auto-proteins is limited by the absence of auto-reactive T cells. The latter are strictly regulated by positive and negative selection eliminating TCRs with a high affinity to self-MHC or self-MHC plus self-antigen. In contrast, B cells are mainly regulated by the presence or absence of antigen-specific T cell help in combination with the soluble antigen. Because of this efficient negative selection of the T cells, there is no need for the B cells to undergo such a strict selection [26]. During B cell maturation, self-reactive B cells are negatively selected in the bone marrow; however, some B cells can escape negative selection when undergoing an additional light chain editing and subsequently leave the bone-marrow into peripheral tissues [27,28,29]. Unless T cell help is provided for a cognate antigen, the thymus-independent (TI) stimulation of B cells can only induce short-lived IgM antibodies without the induction of memory responses. A long-lived IgG production by plasma cells including memory responses always requires antigen-specific T cells and is called thymus-dependent (TD) antibody response [30,31]. The formation of hapten-carrier complexes can bypass this dogma by providing bystander T cell help from the immunogenic carrier to the non-immunogenic but antigenic hapten [12,13,14]. As B and T cells differ in their development, the safety criteria for those cells differ, especially when targeting auto-proteins [32,33]. One important safety requirement for B cell responses when targeting auto-proteins is that the endogenous target protein, in this case endogenous IL-5, should not boost the vaccine-induced anti-IL-5 antibody response. The challenge experiment in mIL-5-VLP-vaccinated mice showed that only a challenge with mIL-5-VLP increased the IL-5-specific antibody titers, whereas a challenge with mIL-5 protein alone did not influence the anti-IL-5 antibody titers. A similar endogenous-mimicry challenge experiment was carried out by Spohn et al. in mice that had been vaccinated with IL-1β-VLP and were challenged with IL-1β, showing comparable results [25]. Although, a limitation of this study is that the challenge experiment was performed in vaccinated mice only, and due to ethical reasons not in vaccinated client-owned horses, we want to highlight the fact that in the absence of a vaccine booster, the anti-IL-5 antibody titers in horses declined. Nevertheless, increased endogenous IL-5 levels are expected throughout the whole IBH season, indicated by enhanced eosinophil levels even at the season end in placebo-vaccinated horses. Taken together, both independent experiments in mice using different auto-proteins linked to VLP strongly indicate that the endogenous IL-5 of the horse is not likely to boost and maintain the IL-5-specific antibody titers. Along that line, a second safety requirement for B cells is the reversibility of the antibody response induced by the vaccine. As described previously, horses that were vaccinated with eIL-5-CuMV_TT_ in the first year, then received a single booster immunization in the second year and one in the third year, had no IL-5-specific antibodies shortly before the booster in the second and the third year, thus showing the reversibility of vaccine-induced anti-IL-5 antibodies [7]. Besides the anti-IL-5 total IgG, the data presented here confirm that the same is true for the IL-5-specific IgG subclasses; without the yearly boost, the IL-5-specific antibody titers decrease to baseline, thus showing the antibody response indeed was reversible. Therefore, the vaccine profile appeared safe from the B cell perspective.

Interestingly, the yearly booster of the vaccine induced similar levels of CuMV_TT_-specific antibodies in the 1st, 2nd, and 3rd year with even a tendency towards higher levels for each year, whilst the IL-5-specific IgG antibodies tended to show lower levels when compared to the initial vaccination year. Nevertheless, at the same time, the horses had significantly decreased levels of eosinophils in the blood and less clinical signs [7], clearly showing the effect of the vaccine. The decreased levels of the anti-IL-5 IgG antibodies in combination with the equally decreased eosinophil levels and reduced clinical signs in follow-up years can be explained by the increased extravasation of the anti-IL-5 antibodies into the target tissue skin and/or the presence of antibody-antigen complexes in the blood, and thus no longer detectable by our ELISA method. Under physiological conditions, the antibody-antigen ICs are cleared from the circulation, involving the innate immune system. As ICs can cause diseases in the kidney and liver when not properly cleared, we measured different parameters in blood indicating the health status of both organs [34,35]. Here, no defects were found for either organ after treatment with the eIL-5-CuMV_TT_ throughout several consecutive vaccination periods. In addition, no effects were seen on the general blood status. Also, as described above, therapeutic antibodies are expected to extravasate to tissues such as the skin, and this will be no longer detectable in the serum. Other studies have shown that the bioavailability of therapeutic antibodies—meaning the potential to extravasate from blood to tissue—depends on several parameters of the antibody, such as the presence of glycostructures and the absence of terminal sialic acids [36].

Furthermore, we suggest that the avidity of the polyclonal antibody response increases from the vaccinations in the first year to the single booster vaccination in the second vaccination year, whereas no further avidity increase is found from the second to the third year, suggesting an affinity maturation of antibodies from the first to the second vaccination year, which does not further increase in the following year (s).

The horse has seven IgG subclasses (IgG1-IgG7) [37] that differ in function, i.e., IgG4/7 and most likely IgG1 protect against viral infections [38,39], while IgG3/5 has been linked to sensitization [40]. The main IgG subclasses following the vaccination with eIL-5-CuMV_TT_ were IgG4/7 and IgG1, which was to be expected as the vaccine appears as a virus to the immune system. We further showed that the IgG4/7 subclass decreases more slowly over time as compared to IgG1, potentially due to the increased bioavailability of the IgG1 subclass. The fact that the anti-IL-5 and the anti-CuMV_TT_ antibody responses are dominated by the same subclasses indicates that both responses were provided the same T cell help, namely the CuMV_TT_-specific T cell help.

The most important T cell safety criterion when vaccinating against a self-protein is that the vaccine does not by any means induce auto-reactive T cells. As discussed above, T cells undergo a very strict negative selection during their development in order to eliminate any auto-reactive T cells. In order to determine if any IL-5-reactive T cells were induced upon the vaccinations, PBMCs from horses vaccinated for one, two, three, four, or five consecutive years were isolated and re-stimulated with *E. coli*-eIL-5, HEK-eIL-5 and CuMV_TT_. Initially, we found low frequencies of IFN-γ^+^ CD4^+^ T cells when the PBMCs from vaccinated horses were in vitro re-stimulated with *E. coli*-eIL-5. However, we also did when re-stimulating with the irrelevant protein *E. coli*-mGM-CSF. The HEK-eIL-5 stimulation did not lead to any IFN-γ production in the CD4^+^ and CD8^+^ T cells, confirming that the IFN-γ production in the above-mentioned cases was due to *E. coli* traces. Additionally, there was no difference between the cells from unvaccinated and vaccinated horses following the HEK-eIL-5 stimulation. At the same time, the HEK-eIL-5-specific IgG was detected in the cell supernatant of resiquimod-stimulated PBMCs and in the plasma of the same horses. This shows that the HEK-eIL-5 protein is relevant and should have been able to stimulate the IL-5-specific peripheral T cells if there had been any. This finding correlates well with the previously published data from Spohn et al., where IL-1β-reactive CD4^+^ T cells were not detected in any of the vaccinated mice [25]. However, here we want to emphasize that we show the lack of peripheral auto-reactive T cells upon vaccination with a self-protein in actual horse patients and over several consecutive years. In contrast with the IL-5 auto-reactive T cells, we do expect to induce CuMV_TT_-reactive T cells following the vaccinations in order to provide T cell help for the IgG responses against both the CuMV_TT_ and the IL-5. Indeed, the PBMCs from vaccinated horses showed significant levels of CuMV_TT_-reactive CD4^+^ and, to a lesser extent, CD8^+^ T cells upon re-stimulation with CuMV_TT_. This was seen both for the IFN-γ production and the proliferation.

## 5. Conclusions

Overall, the data presented here show that vaccination with a cytokine linked to a VLP is a safe way to induce auto-antibodies with the induction of reversible, long-lived, neutralizing, and not-endogen-induced IgG antibody responses, but at the same time without any induction of auto-reactive T cells. The general health status of the horses is not compromised by the induction of auto-antibodies, and the previously published reduction in the clinical signs of IBH leads, on the contrary, to an improved quality of life for the horses. This study comprises the first long-term B and T cell safety analysis upon therapeutic vaccination with a self-antibody-inducing VLP-based vaccine. Performed under field conditions and in a large animal suffering from a real disease, the current study suggests that this vaccination strategy may be a safe replacement for costly monoclonal antibody therapies in companion animals as well as in humans.

## Figures and Tables

**Figure 2 vaccines-08-00213-f002:**
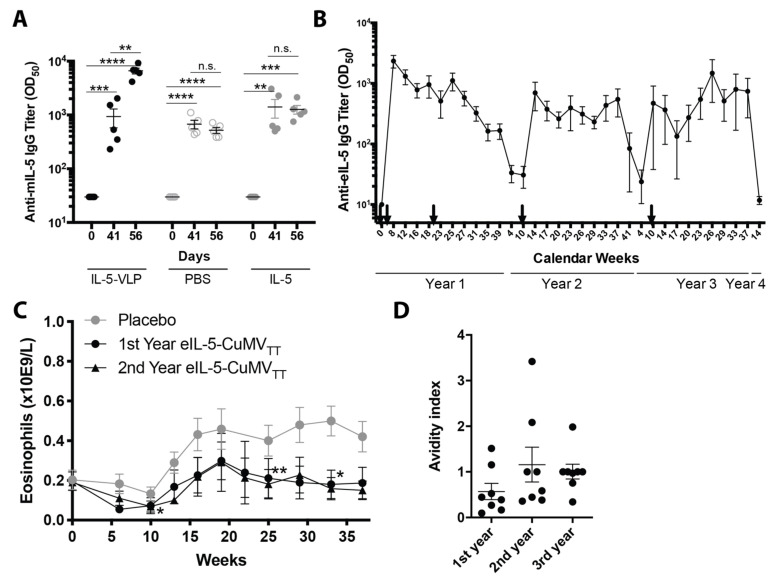
Longitudinal IL-5-specific IgG antibody titer including neutralization and avidity without self-induction. (**A**) Endogenous self-induction. IL-5-specific IgG antibody titers were measured in serum by the ELISA of mice following vaccination with mIL-5-VLP applied on days 0, 14, and 28 and an endogenous-mimicry challenge with either mIL-5-VLP (filled black), PBS (open), or mIL-5 alone (filled grey), applied on day 42. Individual antibody titers are per group (*n* = 5) with mean and SEM. Significances were obtained by a 2-way ANOVA; multiple comparisons with paired t-tests are at different time points within groups. (**B**) Longitudinal anti-IL-5 IgG. Antibody titer reversibility shown by the course of IL-5-specific antibody titers in horses of a 3-year clinical study; IL-5-specific IgG antibody titers (mean +/− SEM) were measured in the serum of horses over three years at different time points. Arrows indicate vaccine injections. The graph contains parts of data of Figure 1 and Figure 2, of Fettelschoss-Gabriel et al. 2019, Allergy. (**C**) Course of eosinophils in the blood during three IBH seasons: placebo-treated season (grey circle, *n* = 13), first year (black circle, *n* = 13), and second year (black triangle, *n* = 11) vaccination season. Part of the data of Figure 2c is published in Fettelschoss-Gabriel et al. 2019, Allergy. The mean and SEM are shown, statistical analysis was performed by the Friedman test. (**D**) Avidity of the IL-5-specific IgG antibodies of eight horses after the yearly booster in the 1st, 2nd, and 3rd year. The avidity index is plotted for individual horses in each year with the mean and SEM; the statistical analysis was performed by the Friedman test (*) followed by a multiple comparison test (*n.s*.). ** *p* < 0.01; *** *p* < 0.001; **** *p* < 0.0001.

**Figure 3 vaccines-08-00213-f003:**
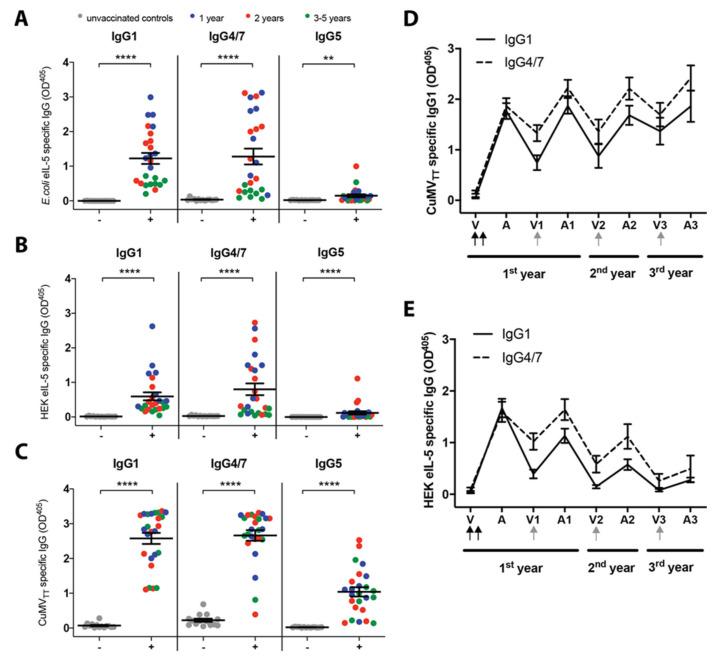
Specific IgG subclasses levels in plasma at the time of the T cell assays. (**A**,**B**,**C**) Subclasses in plasma at the same time as the T cell assays. *E. coli*-eIL-5- (**A**), HEK-eIL-5- (**B**), and CuMV_TT_- specific (**C**) IgG1, IgG4/7, and IgG5 were measured in the plasma of unvaccinated (−) control horses (grey), first (blue), second (red), or third (green) year vaccinated (+) horses by ELISA. OD405 value of the individual horses with the mean and SEM for each group. The Mann–Whitney U test was used to compare the difference between the groups within each subclass. (**D**,**E**) IgG1 and IgG4/7 were measured in the serum of vaccinated horses against CuMV_TT_ (**D**) and HEK-eIL-5 (**E**) by ELISA, mean with SEM. The time points measured: before the initial vaccinations (V), after the initial vaccinations (**A**), at the time of the annual boost at the 1st year (V1), 2–4 weeks after the boost at the 1st year (A1), at the time of the annual boost at the 2nd year (V2), 2–4 weeks after the boost at the 2nd year (A2), at the time of the annual boost at the 3rd year (V3), and 2–4 weeks after the boost at the 3rd year (A3). Black arrows indicate the initial vaccinations and the gray arrows the yearly booster. ** *p* < 0.01; **** *p* < 0.0001.

**Figure 4 vaccines-08-00213-f004:**
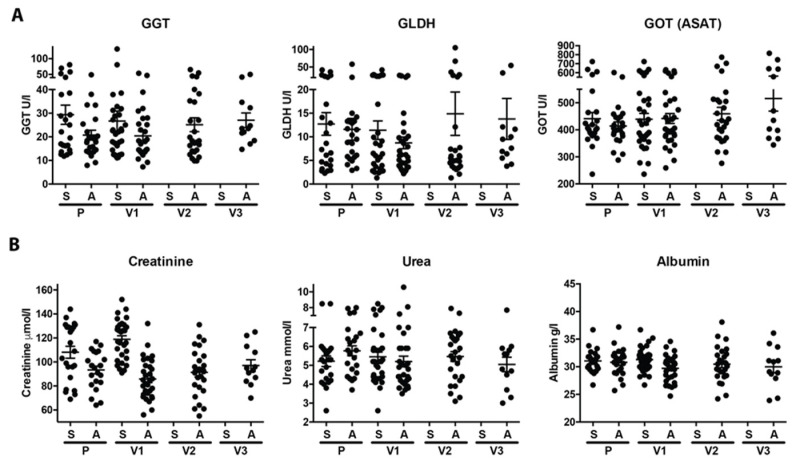
Immune complex disease by liver and kidney parameters. Liver (**A**) and kidney (**B**) parameters were measured in spring (S) and autumn (A) in the placebo year and the 1st (V1), 2nd (V2), and 3rd (V3) treatment year. Values of the individual horses are shown with mean and SEM. Statistical analysis was performed by Kruskal–Wallis with Dunn’s multiple comparison. Statistical significance is only indicated when at least one value was outside of the norm.

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
