# Peer review of "Safety Profile of a Virus-Like Particle-Based Vaccine Targeting Self-Protein Interleukin-5 in Horses"

_vaccines, 2020, doi:10.3390/vaccines8020213_

Round 1

Reviewer 1 Report

The study "Safety profile of a virus-like particle-based vaccine 2 targeting self-protein Interleukin-5 in horses" is an outstanding analysis of therapeutic VLP vaccination against IL5 in horses. Long-term and detailed characterization of immunological profiles in vaccinated and control horses unequivocally demonstrated the induction of strong B-cell responses against IL-5, while no auto-reactive T cells were induced. 

A few typo mistakes can be found below : 

Line 33: “was well tolerated as assessed by serum and cellular biomarkers”

Line 128: “eIL5”

Line 139: “solubilized in 6 M GdmCl”

Line 140: “Qiagen”

Lines 151 and 171: “of re-stimulated equine PBMCs”

Line 199: “overnight”

Line 205: “OD450 titer”

Line 219: “were performed”

Line 271: even though the potential for endogenous IL5 to induce IL-5 auto-antibodies is key in the study by Jonsdottir et al, it has been evaluated in model mice only. Can the authors comment on how they may broaden the interpretation of the results obtained in mice to Iceland horses?

Line 299: when (in weeks) were the eosinophil counts statistically significant between the vaccinated and placebo groups (Figure 2C)?

Author Response

We would like to thank reviewer 1 for his nice comments, the careful reading and valuable input that improved the quality of the manuscript.

Line 33, 128, 139, 140, 151, 171, 199, 219: corrected as suggested. In line 205 we wanted to say OD50 titer, which we defined by the OD50 titer as the reciprocal serum dilution where OD450 reaches the half maximum OD.

Line 271: We agree, this is a very good point and a limitation of our study. As we only included client-owned horses, we thought this experiment was rather classified as unethical as there was no potential benefit for the horse in such an experiment. Also, horses are very sensitive to LPS, and we wanted to prevent injecting E.coli-derived eIL-5 protein intravenously. Nevertheless, the comment of the reviewer is valid and an important one. The reason why we think that similar results might have been achieved in vaccinated horses, is the following: Increased levels of endogenous IL-5 are expected during IBH season indicated by higher eosinophil levels in blood. Of note, placebo-vaccinated horses show enhanced eosinophil levels also at the end of season (Fig. 2C).  However, in absence of a booster, anti-IL-5 antibody titers declined in horses. If endogenously produced IL-5 at the end of IBH-season boost anti-IL-5 antibody titers we would not expect to see almost baseline antibody levels prior to the next seasonal booster. We included that experimental challenge in mice only as a limitation of the study into our discussion and also added our thoughts why we think that endogenous IL-5 in horses does not boost the antibody response in vaccinated horses.

Line 299: We performed a Friedman Test for each week and indicated in the graph when differences were significant. New statistical test is updated in the Figure Legend.

Reviewer 2 Report

The manuscript of Antonia Gabriel and colleagues is clearly written and logically organized. The content is technically sound, and overall the research is well described.  I dont't have any  comments.

Author Response

We would like to thank reviewer 2 for his nice comments.

Reviewer 3 Report

The article by Jonsdottir et al. studies the safety of a virus-like particle-based vaccine against insect bite hypersensivity in horses. The vaccine is constructed by coupling of equine interleukin-5 (IL-5) to Cucumber mosaic virus-like particle (eIL-5-CuMV_TT). The authors showed that eIL-5-CuMV_TT induces specific anibodies and reduces eosinophine levels in blood as well as clinical signs. The experiments are well performed and properly described. In opinion of the reviewer, the article should be published.

The article by Jonsdottir et al. studies the safety of a virus-like particle-based vaccine against insect bite hypersensivity in horses. The vaccine is constructed by coupling of equine interleukin-5 (IL-5) to Cucumber mosaic virus-like particle (eIL-5-CuMV_TT). The authors showed that eIL-5-CuMV_TT induces specific anibodies and reduces eosinophine levels in blood as well as clinical signs. This conclusion is supported by a number of experiments:

1) T cell activation was quantified by the IFN-gamma positive CD4+ and CD8+ T cells following treatment with different stimulants. The vaccine did not induce IL-5 specific T cells, while IL-5-specific B cells were shown to receive bystander T cell help during anti-Il-5 IgG antibody stimulation.

2) The auto-immune safety was checked in mice by three immunizations of mouse IL-5-VLP (mIL-5-VLP) to establish anti-IL-5 antibody titers. Only a challenge with mIL-5-VLP increased IL-5-specific antibody titrs, whereas a challenge with mIL-5 protein alone did not influence anti-IL-5 antibody titers. This is an important issue for B cell responses when targeting auto-proteins.

3)The long-lived anti-IL-5 IgG antibodies were further subclassed and were mostly of the subclass IgG1 and IgG4/ which are known to be invlolved in anti-viral response in horses.

The experiments are well performed and properly described. In opinion of the reviewer, the article should be published.

Author Response

We would like to thank reviewer 3 for his nice comments.